# Cutaneous Adverse Events Following Nemolizumab Administration: A Review

**DOI:** 10.3390/jcm14093026

**Published:** 2025-04-27

**Authors:** Yoshihito Mima, Masako Yamamoto, Ken Iozumi

**Affiliations:** Department of Dermatology, Tokyo Metropolitan Police Hospital, Tokyo 164-8541, Japan

**Keywords:** prurigo nodularis, atopic dermatitis, interleukin-31, T helper 2

## Abstract

Atopic dermatitis (AD) is a chronic inflammatory skin disease characterized by epidermal barrier dysfunction and immune dysregulation, with interleukin (IL)-4, IL-13, and IL-31 recognized as key mediators. Prurigo nodularis (PN) is another chronic inflammatory disorder driven by T helper type 2-mediated inflammation and neural dysregulation, leading to severe pruritus. Nemolizumab, a humanized monoclonal antibody targeting IL-31 receptor A, has been approved for use in the treatment of AD and PN. Clinical trials have demonstrated significant reductions in pruritus and cutaneous symptoms associated with its use. In clinical practice, acute eczema and edematous erythema frequently occur, occasionally necessitating the discontinuation of treatment. Despite these observations, no comprehensive review has examined nemolizumab-associated cutaneous adverse events. This review aimed to examine various cutaneous reactions associated with nemolizumab therapy, including psoriasiform eruptions, AD exacerbation, bullous pemphigoid, drug-induced eruptions, and fungal infections. Potential mechanisms underlying these reactions include T-cell activation due to drug sensitization, immune responses triggered by nemolizumab acting as a hapten, and a relative increase in IL-4 and IL-13 levels following IL-31 inhibition. However, the precise pathophysiological mechanism and risk factors remain unclear, and standardized clinical management guidelines are lacking. Further accumulation of clinical data and immunological research are essential for developing evidence-based strategies to manage these adverse events, ensuring treatment continuity and optimizing patient outcomes.

## 1. Introduction

### 1.1. Atopic Dermatitis and Prurigo Nodularis

Atopic dermatitis (AD) is a chronic, relapsing, non-infectious inflammatory skin disease characterized by persistent pruritus [1]. Clinically, it manifests as eczematous lesions—such as erythema, papules, and exudative eruptions—with variations in distribution and morphology according to the patient’s age (infant, child, or adult) and the degree of skin dryness [2]. Chronic or recurrent inflammation, compounded by repeated scratching, often results in skin thickening and lichenification. Pruritus, a defining feature of AD, often disrupts daily functioning and sleep, leading to insomnia and significantly reduced quality of life (QoL) [3].

AD is associated not only with other allergic conditions such as food allergies, asthma, and allergic rhinitis but also with various systemic comorbidities including hypertension, diabetes, cardiovascular diseases, autoimmune disorders, and psychiatric conditions [4,5]. It is recognized as a multifactorial disease involving complex interactions between genetic predisposition, epidermal barrier dysfunction, immune dysregulation, skin microbiome dysbiosis, and increased exposure to environmental allergens [6]. A family history of atopy is strongly associated with AD, and numerous genes have been implicated in its pathogenesis. Epigenetic modifications, incomplete gene penetrance, and genomic imprinting have been proposed as mechanisms underlying the complex heritability of AD [7,8]. Among the genetic risk factors, loss-of-function mutations in the filaggrin (FLG) gene—which encodes a structural protein essential for maintaining the epidermal barrier—are well-established contributors to AD [9,10]. FLG dysfunction is thought to increase transepidermal water loss, promote skin dryness, enhance antigen presentation by Langerhans cells, and trigger inflammation, thereby exacerbating disease onset and severity [11,12]. AD is primarily driven by type 2 helper T-cell (Th2)-mediated inflammation. Disruption of the skin barrier facilitates antigen penetration and stimulates the release of chemokines (CC chemokine ligand (CCL) 17 and CCL22) and epithelial-derived cytokines such as IL-25, IL-33, and thymic stromal lymphopoietin, which activate type 2 innate lymphoid cells. In AD, CD4+ T cells tend to differentiate into the Th2 lineage, resulting in the overproduction of IL-4, IL-13, IL-5, and IL-31 [13,14,15,16,17]. Importantly, AD exhibits high heterogeneity, with immune profiles varying according to age and ethnicity. In addition to Th2-driven inflammation, the involvement of the Th1, Th17, and Th22 pathways has been discussed, contributing to variability in clinical presentation and treatment responses [18,19,20,21,22,23]. Among the implicated cytokines, IL-13 and IL-31 have been shown to promote the elongation and branching of sensory nerve fibers, thereby intensifying pruritus [24,25]. In AD, the activation of inflammatory cytokines induces pruritus, which triggers scratching and subsequent keratinocyte damage. This, in turn, promotes the release of inflammatory mediators, establishing the well-known “itch–scratch cycle” [26]. Dysbiosis of the skin microbiota also plays a critical role in AD pathogenesis [27]. Commensal microorganisms interact with both innate and adaptive immune systems to maintain immune homeostasis, confer protection against pathogenic organisms, and promote the integrity of the epidermal barrier [28]. However, microbial imbalance—particularly the overgrowth of *Staphylococcus aureus*—disrupts the skin barrier and promotes inflammation by inducing cytokine production, thereby exacerbating the symptoms of AD [29,30]. Consequently, AD is a complex, polygenic disorder influenced by the interplay of genetic predisposition, epidermal barrier dysfunction, immune dysregulation, microbial imbalance, and environmental factors [6].

Prurigo nodularis (PN) is a chronic inflammatory skin disease characterized by intensely pruritic papules, pustules, and nodules measuring from a few millimeters to 2–3 cm in diameter. These lesions typically present in a symmetrical distribution across the limbs and trunk [31,32]. Similar to AD, PN is sustained by the “itch–scratch cycle,” with immune and neural dysregulation playing central roles in disease pathogenesis [33]. The risk factors include eczema, psychiatric disorders, malignancy, hepatic or renal dysfunction, diabetes mellitus, and human immunodeficiency virus infection [34]. Histopathologically, PN is characterized by a Th2-dominant inflammatory infiltrate comprising eosinophils, neutrophils, macrophages, mast cells, and T cells. Like AD, Th2-driven inflammation plays a major role in PN pathogenesis, with elevated levels of IL-4, IL-13, and IL-31 activating peripheral nerves via transient receptor potential channels, thereby eliciting pruritus [26,35,36]. Additional pruritogenic mediators, including tryptase, histamine, prostaglandins, and neuropeptides, further exacerbate this response [37,38,39]. Neural alterations—including increased nerve fiber density and the upregulation of calcitonin gene-related peptide and substance P—further amplify the sensation of itch [38,39]. In PN lesions, IL-31 and oncostatin M (OSM) are upregulated to a greater extent than in AD. These cytokines exert their effects through heterodimeric receptors composed of IL-31 receptor A and OSM receptor β (OSMRβ), directly activating pruriceptive neurons, promoting neuronal growth, and stimulating inflammatory cell activation [40,41]. As in AD, racial and ethnic differences have been considered in PN pathogenesis, underscoring its heterogeneity. PN is increasingly recognized as a multifactorial disease involving not only Th2-driven inflammation but also Th1 and Th17 pathways and neuropeptide dysregulation [42]. Conventional treatments for PN include topical corticosteroids, calcineurin inhibitors, phototherapy, immunosuppressants, anticonvulsants, and antidepressants [43]. With the growing understanding of the role of Th2 cytokines in PN pathogenesis, biologic agents targeting IL-4, IL-13, and IL-31 have been developed as promising therapeutic advances [44,45].

### 1.2. Efficacy of Dupilumab and Nemolizumab for the Treatment of AD and PN

In recent years, biologic agents targeting Th2-associated cytokines such as IL-4, IL-13, and IL-31 have emerged as promising treatment options for AD and PN [46]. This review focuses on dupilumab and nemolizumab, both of which are approved for the treatment of AD and PN. Agents approved only for AD, such as tralokinumab and lebrikizumab, are not discussed herein. Dupilumab, an IL-4 receptor alpha (IL-4Rα) antagonist, was the first biologic approved for use in the treatment of AD and has demonstrated sustained efficacy in Phase III clinical trials [47,48]. In two randomized, placebo-controlled Phase III trials (SOLO 1 and SOLO 2), the efficacy of dupilumab (300 mg every 2 weeks) was evaluated over 16 weeks in adults with moderate-to-severe AD whose disease was inadequately controlled with topical therapy. The primary endpoint was the proportion of patients achieving an Investigator’s Global Assessment (IGA) score of 0 or 1 at week 16. In SOLO 1, this outcome was achieved by 85 patients (38%) in the dupilumab group and only 23 (10%) in the placebo group (*p* < 0.001). In SOLO 2, 84 patients (36%) in the dupilumab group met the primary endpoint compared with 20 patients (8%) in the placebo group (*p* < 0.001). In both studies, a significantly higher proportion of patients receiving dupilumab achieved at least a 75% improvement in the Eczema Area and Severity Index score compared with those receiving the placebo (*p* < 0.001 for all comparisons). Dupilumab was also associated with improvements in pruritus, anxiety or depressive symptoms, and overall QoL. Based on these findings, dupilumab was approved for use in the treatment of AD [47,48]. Its efficacy in PN was demonstrated in two placebo-controlled Phase III trials: LIBERTY-PN PRIME and PRIME2 [49]. Patients received 300 mg of dupilumab (with a 600 mg loading dose) or a placebo via subcutaneous injection every 2 weeks for 24 weeks. The concomitant use of low-to-mid-potency topical corticosteroids or topical calcineurin inhibitors was permitted. At week 24, 60.0% of patients in the PRIME trial receiving dupilumab achieved a ≥4-point improvement in Peak Pruritus Numerical Rating Scale (PP-NRS) scores compared with 18.4% in the placebo group (*p* < 0.001). Furthermore, 48.0% of patients in the PRIME trial and 44.9% in PRIME2 achieved a Prurigo Nodularis Investigator’s Global Assessment (PN-IGA) score of 0 or 1 in the dupilumab groups, compared with 18.4% and 15.9%, respectively, in the placebo groups (*p* < 0.001) [49]. These results led to regulatory approval, and subsequent real-world studies have confirmed the clinical effectiveness of dupilumab in PN [49,50]. Collectively, these findings underscore the involvement of Th2 cytokines—particularly IL-4 and IL-13—in PN pathogenesis and suggest that dupilumab may improve symptoms by disrupting the itch–scratch cycle [26,35,36,49,50].

Nemolizumab is a humanized monoclonal antibody that targets the interleukin-31 receptor A (IL-31RA) [51]. Its efficacy in AD was demonstrated in a 16-week, double-blind, Phase III clinical trial involving Japanese patients with moderate-to-severe pruritus who had shown an inadequate response to topical therapies. The patients were randomly assigned in a 2:1 ratio to receive subcutaneous injections of nemolizumab (60 mg) or placebo every 4 weeks for 16 weeks, in combination with standard topical treatment. At week 16, the mean percent changes in visual analogue scale (VAS) scores for pruritus were −42.8% in the nemolizumab group and −21.4% in the placebo group (difference: −21.5 percentage points; 95% confidence interval: −30.2 to −12.7; *p* < 0.001). The mean percentage changes in the Eczema Area and Severity Index (EASI) scores were −45.9% in the nemolizumab group and −33.2% in the placebo group. The proportions of patients achieving a Dermatology Life Quality Index (DLQI) score of ≤4 were 40% in the nemolizumab group and 22% in the placebo group. Similarly, the proportions of patients with an Insomnia Severity Index score of ≤7 were 55% and 21% in the nemolizumab and placebo groups, respectively. These results led to the approval of nemolizumab as a novel treatment for AD, given its significant antipruritic effects compared with a placebo [52]. The efficacy of nemolizumab for PN was demonstrated in the OLYMPIA2 trial, a Phase III randomized, placebo-controlled study. Patients received either 30 mg nemolizumab monthly (with an initial loading dose of 60 mg) or a placebo for 16 weeks. Unlike the PRIME1 and PRIME2 trials of dupilumab, topical treatments for PN were not permitted during this study. At week 16, 56.3% of patients receiving nemolizumab achieved a ≥4-point improvement in the PP-NRS score compared with 20.9% of those in the placebo group (*p* < 0.001). Additionally, 37.7% of patients in the nemolizumab group achieved a PN-IGA score of 0 or 1 compared with 11.0% of those in the placebo group (*p* < 0.001) [53]. IL-31 plays a key role in the pathogenesis of PN by promoting fibrosis and neuroinflammation. By blocking IL-31 signaling, nemolizumab may help prevent disease progression [53,54]. Based on the results of this randomized, double-blind, placebo-controlled Phase III trial—demonstrating significant improvements in NRS scores at week 16, higher proportions of patients achieving a ≥4-point improvement in NRS scores, and higher PN-IGA 0/1 response rates—nemolizumab received regulatory approval and was launched in 2024 [53]. Notably, nemolizumab has also demonstrated a rapid onset of action, with significant improvements in pruritus scores and sleep disturbances observed within days of administration [54]. In a long-term clinical trial evaluating the efficacy of nemolizumab for PN over a 68-week period, sustained improvements in both PP-NRS and PN-IGA scores were observed, with no decline in effect during continued treatment for more than one year [55,56].

### 1.3. Adverse Events of Dupilumab and Nemolizumab

In the SOLO1 and SOLO2 trials, which evaluated the efficacy and safety of dupilumab for AD, serious adverse events (AEs) were shown in seven patients (3%) and four patients (2%), respectively. Treatment discontinuation due to AEs occurred in four patients (2%) and two patients (1%), respectively. The most common AEs (≥10%) included AD exacerbation and nasopharyngitis, whereas other reported events included herpesvirus infections, headache, psoriasis, arthritis, vitiligo, alopecia areata, rosacea-like folliculitis, and conjunctivitis [48,49]. In Phase III clinical trials for PN, PRIME1 and PRIME2, serious AEs were observed in five patients (6.7%) and two patients (2.6%), respectively; however, none of the patients discontinued treatment during the 24-week study period. Nasopharyngitis and headache were observed in ≥5% of patients, with additional reports of coronavirus disease 2019 (COVID-19), skin infections, conjunctivitis, and herpesvirus infections [49]. These findings indicate that dupilumab is well tolerated in both AD and PN, with a low rate of treatment discontinuation due to AEs, supporting its stable long-term use. Post-marketing surveillance data further corroborate the favorable safety profile of dupilumab, with very few cases leading to discontinuation [50].

A clinical trial evaluating nemolizumab for AD reported the occurrence of AEs in 71% of patients. Serious AEs developed in three patients (2%), and treatment discontinuation due to AEs was also observeded in three patients (2%). The most common AEs included AD exacerbation in 33 patients (23%), nasopharyngitis in 18 patients (13%), and skin infections in 10 patients (7%). Additional AEs included elevated creatine kinase levels and acneiform eruptions [52]. In a 16-week, Phase III placebo-controlled trial investigating nemolizumab for PN, AEs occurred in 61% of patients, with serious AEs observed in four patients (2.2%) and treatment discontinuation in four patients (2.2%). The most commonly reported AEs included viral and skin infections in 39 patients (21%), gastrointestinal symptoms in 11 patients (6%), and AD exacerbation in 10 patients (5.5%) [53]. In a 68-week extension study, over 90% of patients experienced at least one AE, but they were mostly mild. Moreover, the treatment discontinuation rate remained low, at 4–6%. The most frequently observed AEs included newly developed or worsened eczema (20%), COVID-19 (15%), urticaria (5%), nonspecific erythema (5%), contact dermatitis (5%), acne (5%), and nasopharyngitis (5%) [54]. Compared with the 16-week study, the incidence of mild AEs tended to increase during the 68-week treatment period; however, the proportion of patients discontinuing treatment remained consistent at 4–6%, and the incidence of serious AEs did not increase over time. These findings suggest that nemolizumab is a well-tolerated treatment and remains safe for long-term use in patients with PN [53,54].

### 1.4. Correlation Between Nemolizumab and T Helper 2 or 17 Inflammation

IL-31 is considered a Th2-type cytokine, and nemolizumab demonstrated significant efficacy in AD in a phase III double-blind trial, with marked improvements in VAS, EASI, and DLQI scores at week 16 [52]. In PN, the OLYMPIA2 trial confirmed significant improvements in PP-NRS and PN-IGA scores at week 16 [53]. These findings support the central role of IL-31 in the pathogenesis of both PN and AD. However, AEs such as AD exacerbation and non-specific edematous erythema have been observed in over 10–20% of patients following nemolizumab treatment [55], suggesting that IL-31 inhibition may not exert uniformly anti-inflammatory effects. Additionally, a transient elevation in serum activation-regulated chemokine (TARC) levels was observed in approximately 5% of patients [56], indicating the potential involvement of IL-31 in a negative feedback loop regulating other Th2 cytokines, such as IL-4 and IL-13 [55,56]. IL-31 signals through a heterodimeric receptor composed of IL-31RA and OSMRβ. Notably, aOSMRβ also forms a receptor complex with gp130 to mediate oncostatin M (OSM) signaling. In IL-31RA-deficient mice, OSM-dependent cytokine production is enhanced, suggesting that the IL-31RA blockade may activate OSM signaling and thereby exacerbate Th2-type inflammation [56,57,58]. Furthermore, IL-31 knockout mice exhibit increased cutaneous expression of IL-4 and IL-13, and the transient elevation of TARC levels observed after nemolizumab treatment may be attributed to this compensatory mechanism [56,57,58,59]. Thus, paradoxically, IL-31 inhibition may contribute to the upregulation of Th2-mediated inflammation.

Conversely, IL-31 inhibition may also influence Th17-related inflammatory pathways. Cutaneous immune homeostasis is regulated through a dynamic balance between T-cell subsets, including Th1, Th2, Th17, and regulatory T cells. The suppression of one subset may lead to the compensatory activation of another. Therefore, the inhibition of IL-31—a key Th2 cytokine—may attenuate Th2-driven inflammation while promoting Th17 responses, leading to an imbalance in immune regulation [60,61].

Thus, compared with dupilumab, nemolizumab is currently associated with a higher incidence of cutaneous AEs, such as eczema and the exacerbation of AD [47,48,49,50,51,52,53]. However, to the best of our knowledge, no published comprehensive review has specifically examined cutaneous AEs associated with the administration of nemolizumab. Therefore, this review aimed to elucidate the clinical features and management strategies for cutaneous-related adverse effects arising from nemolizumab therapy.

## 2. Review of Nemolizumab-Associated Cutaneous Adverse Events

The incidence and types of cutaneous AEs reported in clinical trials of nemolizumab for AD, including pediatric AD, and PN, are summarized in Table 1 [52,53,54,56,62,63,64]. In trials with relatively short evaluation periods of 12–16 weeks, the overall AE incidence ranged from 60% to 70% [52,53,63], exceeded 80% in 24-week trials [62], and rose to over 90% in longer-term studies of 68 weeks [54,56,64]. The most frequently reported cutaneous AEs include new-onset eczema and the exacerbation of AD, occurring in 10–30% of participants. Other cutaneous-related AEs such as contact dermatitis and non-specific erythema were observed in approximately 6–8% of patients. Neurodermatitis, urticaria, and acne were also reported frequently. However, considerable variability exists in how AEs are classified across studies. For example, some trials differentiate between general eczema and AD exacerbation, whereas others may classify acne under folliculitis. These inconsistencies in classification make it challenging to uniformly summarize or compare cutaneous AEs. Additionally, pediatric AD trials often report a higher incidence of acne, likely due to age-related factors. Moreover, certain AEs, such as asteatotic eczema, contact dermatitis, and urticaria, may occur independently of nemolizumab treatment. Therefore, the extent to which these events can be directly attributed to the drug itself remains unclear.

Nemolizumab has been approved for use in the treatment of AD and PN, and clinical trials have reported various AEs, as summarized in Table 1. This review focuses on cutaneous AEs that have been highlighted in case reports as potentially being associated with nemolizumab administration—namely, psoriasiform eruptions, bullous pemphigoid (BP), the exacerbation of AD, non-specific erythema, and fungal skin infections. The possible underlying mechanisms through which these reactions may arise following nemolizumab treatment were also discussed (Figure 1).

### 2.1. Psoriasis-like Eruptions

Psoriasis is a chronic inflammatory skin disease characterized by cycles of relapse and remission, driven by complex immunological interactions. Although it was previously considered a Th1-mediated disorder, recent studies have highlighted the pivotal role of the Th17 pathway in its pathogenesis [65,66,67]. Key cytokines involved include IL-17, IL-21, IL-22, interferon-γ, tumor necrosis factor-α, IL-6, IL-20, and IL-23, which are produced by T cells, dendritic cells, and keratinocytes [68]. Psoriasiform eruptions resemble psoriasis in appearance and are typically triggered by identifiable factors such as medications or infections [69]. Due to their clinical presentation, psoriasiform eruptions often require differentiation from pityriasis-rosea-like eruptions. The latter is believed to be associated with the reactivation of human herpesvirus 6 and 7 and is histopathologically characterized by vacuolar degeneration and lymphocytic and eosinophilic infiltration in the subepidermal region—features that help distinguish them from psoriasiform eruptions [70,71]. Masuda et al. reported a patient who developed a psoriasiform eruption after two doses of nemolizumab for AD, despite marked improvement in pruritus. A histological analysis revealed increased CD3-positive cells, as well as retinoic acid-related orphan receptor C- and GATA-3-positive cell counts, suggesting post-treatment Th17 activation. The eruption was localized and successfully managed with continued nemolizumab therapy and the adjustment of topical treatment [72]. Although the exact mechanism is unclear, IL-31 inhibition may suppress Th2 inflammation and trigger a compensatory Th17 response. Similar eruptions have been seen with dupilumab, likely due to the activation of the Th17/IL-23 axis after Th2 suppression [57,72,73]. In vitro studies suggest that Th2 inflammation suppresses IL-23 production by dendritic cells [57]; thus, the IL-31 blockade may disrupt this balance, promoting Th17-driven psoriasiform lesions.

### 2.2. Bullous Pemphigoid

BP is the most common autoimmune subepidermal blistering disease that primarily affects older adults. It presents with urticarial erythema and tense blisters caused by autoantibodies targeting BP180 and BP230 at the dermal–epidermal junction [74]. These antibodies activate complement, recruit inflammatory cells, and initiate proteolytic cascades, leading to blister formation [75,76,77]. Known triggers for BP include dipeptidyl peptidase-4 inhibitors, diuretics, checkpoint inhibitors, viral infections, malignancies, and neurological disorders [78,79]. BP is closely associated with Th2-mediated inflammation, as evidenced by elevated levels of IL-4, IL-5, IL-6, IL-10, and IL-13 in serum, tissue, and blister fluid [80,81,82]. The efficacy of dupilumab in treating BP further supports the involvement of Th2 pathways [83,84]. Ishikawa et al. reported the occurrence of BP after two doses of nemolizumab in a patient with AD. The patient presented with elevated eosinophil and thymus and TARC levels, subepidermal blisters, eosinophilic dermal infiltration, and linear immunoglobulin G/complement component 3 deposition—findings consistent with BP. The patient responded to prednisolone after the discontinuation of nemolizumab [85]. Similarly, Masuyuki et al. reported a patient who required multiple therapies before achieving disease control with dupilumab monotherapy [86]. Transient TARC elevation after nemolizumab treatment suggests that IL-31 provides negative feedback on IL-4/13 [55,56,85,86]. Disrupting this feedback may paradoxically enhance Th2 inflammation and trigger BP [55,56,80,81,82,83,84,85,86]. Further research is needed to clarify these mechanisms and better understand nemolizumab-related immune responses.

### 2.3. Exacerbation of Atopic Dermatitis

AD is a chronic, relapsing eczematous skin disease with a complex presentation, driven by epidermal barrier dysfunction, impaired innate immunity, and a Th2-dominant response mainly involving IL-4 and IL-13 [1,2,3,4,5,6,7,8,9,10,11,12,13,14,15,16,17,18,19,20,21,22,23,24,25,26,27,28,29,30]. Kamada et al. reported a paradoxical exacerbation of AD following nemolizumab treatment, characterized by worsening edematous erythema and eczematous lesions. A histopathological analysis revealed an increased number of dendritic cells and the activation of CD4-positive T cells, findings suggestive of a flare of the underlying disease rather than a drug-induced eruption [87,88]. Similar exacerbations of AD have been observed in clinical trials involving nemolizumab administration. Although treatment discontinuation is recommended in patients with severe conditions, mild exacerbations may be managed by the continuation of nemolizumab in combination with topical or systemic corticosteroids [55,56,87].

### 2.4. Non-Specific Drug-Induced Eruptions Such as Edematous Erythema or Acute Eczema

Kamada et al. reported a case of a patient with AD who developed multiple keratotic, erythematous papules on the trunk and extremities following treatment with nemolizumab [89,90]. A histopathological examination revealed increased CD11c⁺ and HLA-DR⁺ dendritic cell counts and the activation of CD8⁺ T cells, resulting in a diagnosis of a drug-induced eruption. Another study reported the development of erythema multiforme-like eruption three weeks after the initial administration of nemolizumab in a patient with PN [91]. In addition to systemic drug-induced eruptions, localized and non-specific skin reactions—such as acute eczema and edematous erythema—have frequently been observed [56,92]. In an observational study involving 25 patients with AD, cutaneous AEs occurred in 80% of patients, most commonly after the first or second dose of nemolizumab [92]. Mild reactions were managed by continuing nemolizumab therapy in combination with intensified topical treatment, whereas more severe symptoms required switching to alternative biologics, such as dupilumab or lebrikizumab [37]. Similar eruptions have also been observed in patients with PN following the first or second dose of nemolizumab (Figure 2 and Figure 3). Mild symptoms were managed with intensified topical corticosteroids; however, in patients with more severe symptoms or in patients who declined continued treatment, nemolizumab was either discontinued or temporarily withheld. In some instances, therapy was switched to alternative biologics such as dupilumab. When the use of alternative biologics was not feasible due to financial constraints, a short course of oral corticosteroids was administered. These cutaneous reactions may be mediated by drug-specific T-cell activation or hapten formation by endogenous proteins [93,94]. Furthermore, the compensatory upregulation of IL-4 and IL-13 following IL-31 blockade may contribute to the development of non-specific drug-induced eruptions [56,57,58,59,92].

### 2.5. Fungal Infection

A patient with concomitant AD and PN developed Malassezia folliculitis on the anterior chest following the initial administration of nemolizumab (Figure 4). The eruption was mild and resolved within three weeks with topical ketoconazole, allowing the continuation of nemolizumab therapy. The skin microbiome is composed not only of bacteria such as *Staphylococcus* species but also fungi such as *Malassezia*, both of which are considered part of the normal skin flora [95]. Anti-Th2 therapies such as dupilumab may reduce the abundance of Staphylococcus, potentially allowing commensal fungi to overgrow [96]. Similarly, nemolizumab may shift the microbial balance by suppressing Th2 responses, promoting fungal predominance [95,96]. Th17 activation—linked to both antifungal immunity and psoriasiform eruptions—may further contribute to this [60,61,97]. In dupilumab-associated facial erythema, Malassezia overgrowth and heightened Th17 responses have been suggested [98]. While fungal infections depend on various factors, these findings indicate that nemolizumab may promote fungal infections through microbiome alterations and Th17 activation.

### 2.6. Others (Urticaria, Contact Dermatitis, and Acne)

In previous clinical trials, urticaria, contact dermatitis, and acne have been reported as common cutaneous AEs [52,53,54,56,62,63,64]. Urticaria, a Th2-driven condition, may paradoxically develop after IL-31 inhibition by nemolizumab due to disrupted negative feedback and the compensatory upregulation of Th2 cytokines [99]. In contrast, contact dermatitis is a Th1/Th17-mediated delayed-type hypersensitivity reaction, and Th2 suppression by nemolizumab may enhance Th1/Th17 activity, increasing susceptibility [100,101]. Acne has been linked to alterations in the skin microbiome, particularly the balance of resident *Staphylococcus* species [102,103]. The inflammatory control of AD or PN with nemolizumab may reduce *Staphylococcus* overgrowth, leading to microbial shifts that favor the proliferation of acne-related microorganisms. This microbial imbalance may unmask or exacerbate acne symptoms [102,103]. Thus, immunological shifts and alterations in the skin microbiome following the administration of nemolizumab may contribute to the development of urticaria, contact dermatitis, and acne [101,102,103]. However, these conditions may also arise independently due to external stimuli or endogenous factors. Therefore, further investigation is warranted to clarify the extent to which nemolizumab contributes to their onset.

## 3. Discussion

Nemolizumab has been approved for use in the treatment of AD and PN, with clinical trials demonstrating its efficacy in improving both skin lesions and pruritus [52,53,54,55,56]. In real-world clinical settings, various cutaneous AEs—including asteatotic eczema, edematous erythema, and acute eczema—have frequently been reported [52,53,54,55,56]. Among case series involving patients with AD and PN, the most commonly observed AE is the appearance of eczematous eruptions or exacerbation of AD, occurring in approximately 20–30% of patients. Other relatively frequent events include erythema, acne, urticaria, and contact dermatitis, each reported in fewer than 10% of patients [52,53,54,55,56]. Overall, cutaneous AEs have been observed in approximately 30–50% of patients receiving nemolizumab therapy [52,53,54,55,56]. By contrast, an observational study involving 25 Japanese patients with AD reported a markedly high incidence of cutaneous AEs, with 80% of participants developing skin reactions—most of which occurred after the first or second dose of nemolizumab [95]. This discrepancy may be attributed to several factors, such as the reduced use of topical therapies following pruritus improvement, ethnic differences in immune response, or variations in the definitions and reporting criteria for AEs. Notably, approximately 30% of patients in this Japanese cohort discontinued nemolizumab due to these adverse effects [95]. Despite these findings, only a few systematic evaluations have focused on the spectrum of cutaneous AEs associated with nemolizumab. To address this gap, the present review aimed to clarify the full range of cutaneous-related adverse effects linked to nemolizumab treatment and to explore appropriate strategies for their clinical management.

The cutaneous AEs associated with nemolizumab include psoriasiform eruptions, the exacerbation of AD, BP, non-specific drug-induced eruptions, fungal infections, urticaria, acne, and contact dermatitis [57,59,65,66,67,68,69,70,71,72,73,74,75,76,77,78,79,80,81,82,83,84,85,86,87,88,89,90,91,92,93,94,95,96,97,98,99,100,101,102,103,104]. Although the onset of fungal infections and acne may be influenced by external factors, emerging evidence suggests that nemolizumab-induced alterations in the cutaneous microbiome—along with a shift toward Th17-dominant inflammation—may predispose patients to infections caused by organisms such as *Cutibacterium acnes* and *Malassezia* spp. [95,96,97,98,102,103]. Furthermore, psoriasiform eruptions and contact dermatitis may result from the suppression of Th2 cytokines due to IL-31 inhibition, which can lead to the compensatory activation of Th1 and Th17 pathways [57,65,66,67,68,69,70,71,72,73,100,101]. Conversely, IL-31 regulates IL-4 expression. Therefore, its inhibition may paradoxically lead to IL-4 upregulation, potentially triggering Th2-related disorders such as AD, BP, and urticaria [74,75,76,77,78,79,80,81,82,83,84,85,86,87,88,89,90,91,92,93,94,99]. IL-31 and OSM signal through a heterodimeric receptor composed of IL-31RA and OSMRβ. The inhibition of IL-31 may lead to the hypersensitization of OSM signaling, thereby further enhancing Th2-driven inflammation. Indeed, clinical trials in AD have reported transient increases in serum TARC levels persisting for up to 32 weeks in approximately 5% of patients, suggesting the paradoxical activation of Th2-driven inflammation in certain individuals. By contrast, transcriptomic analyses of patients with PN have shown decreased IL-13 expression after 12 weeks of treatment, demonstrating a Th2-suppressive effect in other contexts [104]. These findings underscore the heterogeneous immune responses to nemolizumab, whereby some patients with AD or PN experience clinical improvement, whereas others may develop new or worsening eczematous lesions. The pathogenesis of both conditions involves complex interactions among the Th1, Th2, and Th17 pathways. Differences in individual immune responses to IL-31 inhibition may be influenced by genetic background, cytokine profiles, and overall immune homeostasis [104,105]. Consequently, the effect of IL-31 inhibition on Th2-driven inflammation remains controversial and warrants further investigation [55,56,57,58,59,104]. Edematous erythema and erythematous papules may develop not only as manifestations of type IV hypersensitivity reactions—such as drug-specific T-cell sensitization or hapten formation—but also as a result of cytokine imbalances, particularly the upregulation of IL-4 [57,89,90,91,92,93,94]. Moreover, the severity of drug-induced eruptions has been shown to correlate with peripheral eosinophil counts [106], and elevated eosinophil levels have been observed in some patients receiving nemolizumab [105], suggesting the potential role of eosinophilic inflammation in the development of these adverse reactions. However, as most cutaneous AEs occur after the first or second dose and are rarely observed after the third dose, these eruptions are likely driven by [52,53,54,55,56,57,58,59,60,61] cytokine imbalances induced by IL-31 inhibition rather than by delayed-type hypersensitivity [92]. In AD or PN with prominent erythema and barrier dysfunction—where IL-4 and IL-13 are likely elevated—nemolizumab may transiently increase TARC and activate Th2 responses, raising the risk of Th2-related AEs. In contrast, cases with severe lichenification or pruritus hypersensitivity—where IL-31 is more involved and IL-4/IL-13 levels are lower—may be less prone to such AEs. As risk factors remain unclear, further case accumulation and subgroup analysis are needed to identify high-risk patients [52,53,54,55,56,57,58,59,60,61].

The recommended management approach for mild to moderate cutaneous AEs involves continuing nemolizumab therapy while intensifying topical treatments or administering a short course of systemic corticosteroids [52,53,54,56,62,63,64,92]. In a 68-week clinical trial of nemolizumab, rates of treatment discontinuation due to AEs remained under 10%, indicating that most patients could be effectively managed with appropriate topical therapy or short-term oral prednisolone [52,53,54,56,62,63,64,92]. If symptoms persist, extending the dosing interval of nemolizumab may be considered. In patients with more severe or refractory AEs, switching to an alternative biologic agent—such as dupilumab or lebrikizumab—may be an effective strategy, particularly as IL-4/IL-13 signaling is likely implicated in these cutaneous manifestations [92]. However, due to financial constraints, some patients may be unable to access clinically appropriate biologics, such as dupilumab or lebrikizumab, and may need to rely on topical agents or systemic corticosteroids. Currently, there are no standardized criteria for classifying cutaneous AEs as mild, moderate, or severe, and nemolizumab trials lack clear benchmarks based on skin involvement or pruritus severity. At our institution, severity is evaluated with the patient’s opinion. AEs are considered mild if treatment continues with topical or oral steroids, moderate if they lead to a request for dosing interval extension, and severe if they result in treatment discontinuation. Standardized criteria based on skin area and pruritus intensity are needed for more consistent assessment. Based on data from clinical trials, published case reports, and our institutional experience, a management algorithm was developed (Figure 5) to guide clinical decisions when cutaneous AEs arise during nemolizumab therapy. Although this flowchart may serve as a practical reference, engaging in shared decision making with each patient is essential to determining the most appropriate treatment strategy tailored to their clinical status and personal circumstances. Importantly, a significant proportion of patients may be forced to discontinue nemolizumab before achieving their therapeutic goals, such as pruritus relief or skin improvement, due to the associated financial burden. The proposed algorithm in Figure 5 reflects our current opinion and experience; however, standardized clinical guidelines for managing nemolizumab-associated cutaneous AEs have yet to be established. Further case accumulation and evidence-based evaluation are needed to determine when treatment discontinuation should be considered and to develop consensus-driven strategies for the optimal management of these events.

The primary limitation of this review lies in its reliance on data derived from existing clinical trials and case reports. Due to the high cost of nemolizumab as a biologic agent, no studies to date have reported the use of patch testing in association with its use. Furthermore, allergy testing, such as drug-induced lymphocyte stimulation tests (DLSTs), is often impractical in routine clinical practice. As a result, it remains unclear whether the cutaneous reactions observed following the administration of nemolizumab represent conventional type IV hypersensitivity reactions (i.e., drug eruptions) or erythematous responses driven by cytokine shifts, such as IL-4 upregulation secondary to IL-31 inhibition. However, both patch testing and DLST carry substantial risks of false-positive and false-negative results [107,108], and the inability to perform these tests does not necessarily impose a significant limitation on therapeutic decision making or lead to major disadvantages for patients. Accordingly, the indication for such testing should be determined on a case-by-case basis, based on the clinical context and the patient’s preferences.

As a relatively new therapeutic agent, nemolizumab remains insufficiently characterized in terms of the incidence of cutaneous AEs, the profiles of high-risk patients, and the underlying pathophysiological mechanisms. Moreover, no standardized management algorithms have been established to date; in clinical practice, treatment is often based on a trial-and-error approach. Despite its therapeutic potential, a considerable number of patients discontinue nemolizumab due to AEs, even when the drug would otherwise represent the most suitable therapeutic option; this represents a significant clinical concern. Given these challenges, developing evidence-based management strategies to support the safe and effective continuation of nemolizumab therapy remains an urgent priority. Achieving this will require the accumulation of additional clinical cases and continued advances in basic and translational research to clarify the immunopathogenesis of nemolizumab-associated cutaneous AEs.

## Figures and Tables

**Figure 1 jcm-14-03026-f001:**
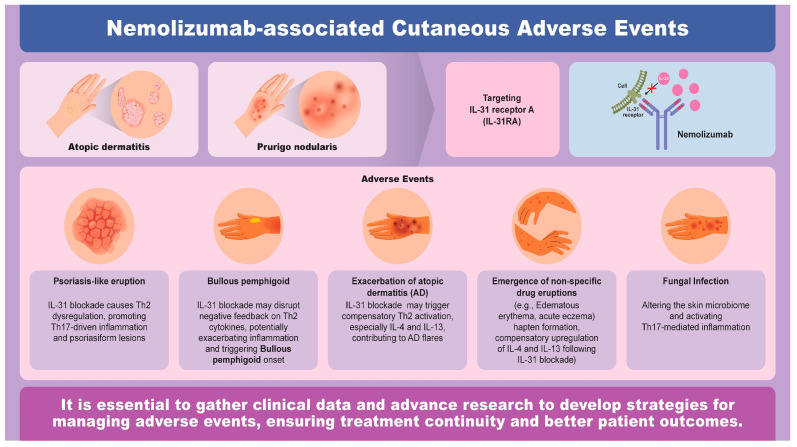
Overview of nemolizumab-induced cutaneous adverse events.

**Figure 2 jcm-14-03026-f002:**
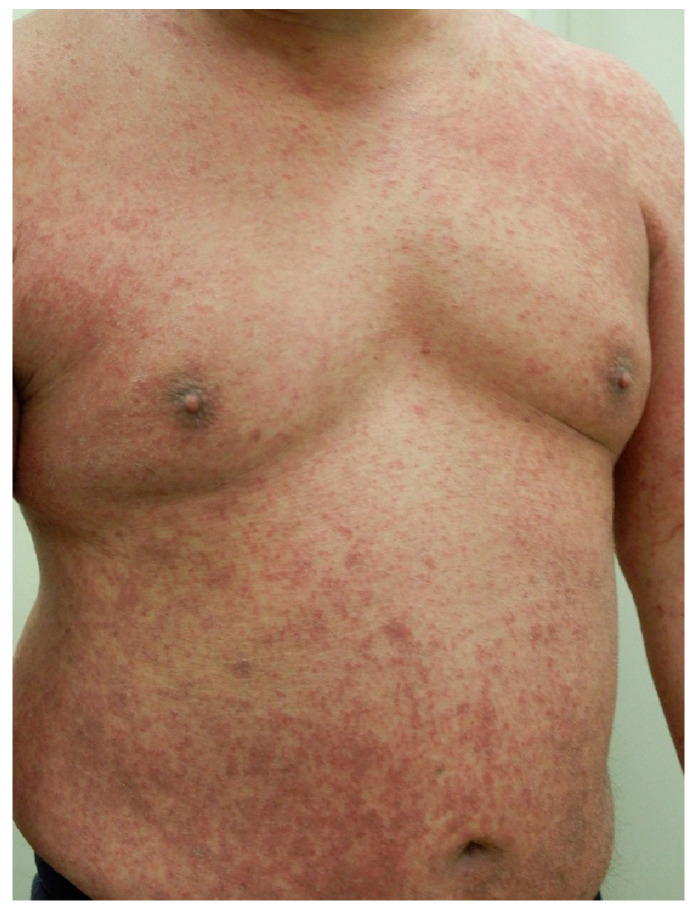
A 50-year-old man with treatment-resistant prurigo nodularis that had been managed with topical corticosteroids. Due to severe pruritus, nemolizumab was initiated at a dose of 60 mg, followed by 30 mg monthly. After the second dose, although the pruritus had begun to improve, the patient developed acute eczema with widespread skin involvement, prompting the discontinuation of nemolizumab. Transition to dupilumab was initially deferred due to financial constraints. Instead, a course of oral corticosteroids combined with intensified topical therapy was administered, resulting in improvement within two weeks. The patient continued treatment with topical corticosteroids thereafter; however, residual pruriginous nodules and pruritus persisted in the lower limbs. Ultimately, treatment was transitioned to dupilumab, which is currently ongoing.

**Figure 3 jcm-14-03026-f003:**
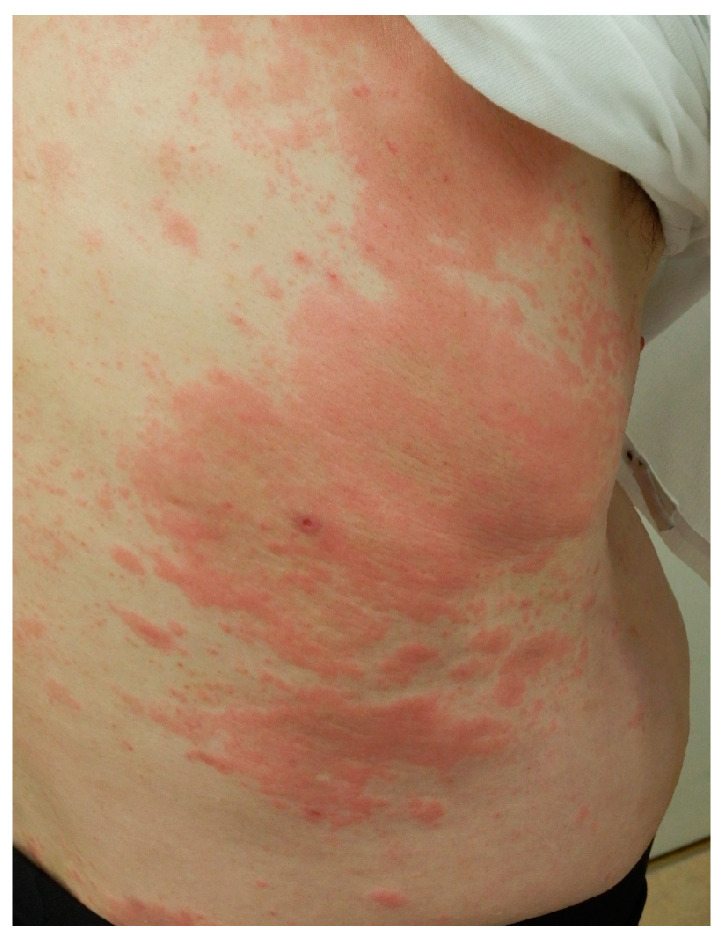
A 66-year-old man with treatment-resistant prurigo nodularis that had been managed with topical corticosteroids. Due to severe pruritus, nemolizumab was initially administered at a dose of 60 mg, followed by 30 mg monthly. After the second dose, localized edematous erythema developed on the trunk. The eruption resolved within one week with the application of clobetasol propionate ointment. The topical regimen was subsequently stepped down to betamethasone valerate, and nemolizumab treatment was continued. The pruritus and skin lesions have since remained well controlled.

**Figure 4 jcm-14-03026-f004:**
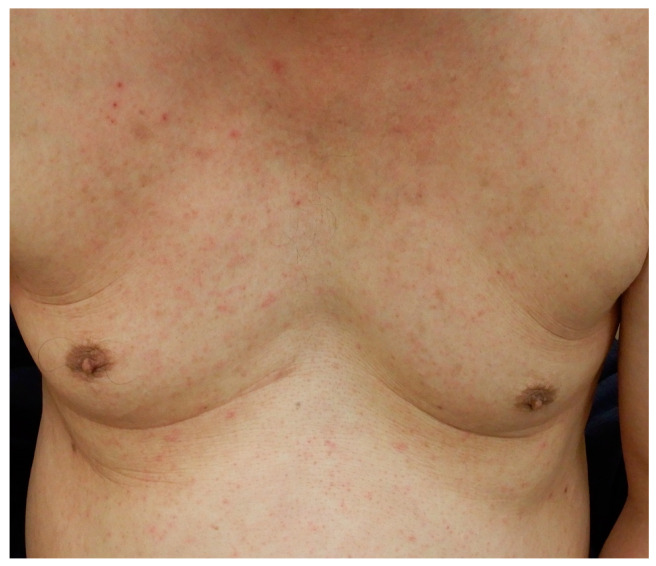
A 77-year-old man with treatment-resistant prurigo nodularis, who had received topical corticosteroid therapy. Due to severe pruritus, nemolizumab was initiated at a dose of 60 mg, followed by 30 mg monthly. Pruritic erythematous papules developed on the chest, and a fungal examination revealed the presence of hyphae and spores, leading to a clinical diagnosis of *Malassezia* folliculitis. Symptoms resolved within three weeks after topical ketoconazole therapy. Nemolizumab treatment was continued, and both the pruritus and skin lesions remained well controlled.

**Figure 5 jcm-14-03026-f005:**
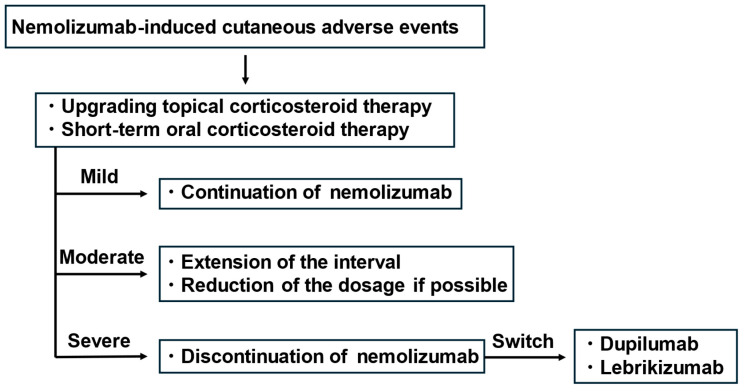
Proposed strategy for managing nemolizumab-induced adverse events, based on clinical trials, case series, and our clinical experience. Final decisions should be made through shared decision making with patients.

**Table 1 jcm-14-03026-t001:** Incidence and types of cutaneous adverse events reported in clinical trials of nemolizumab.

Article	Disease	Amount	Duration	Number	Any AEs	Eczema or AD Exacerbation	Erythema	Neurodermatitis	Acne	Urticaria	Contact Dermatitis
Kabashima K et al. [57]	AD	60 mg	68 W	298	281 (94.3%)	75 (25.2%)	N/A	N/A	22 (7.4%)	24 (8.1%)	26 (8.7%)
Silverberg JI et al. [63]	AD	30 mg	24 W	57	47 (82.5%)	17 (29.9%)	N/A	N/A	N/A	N/A	N/A
Igarashi A et al. [64]	Pediatric AD	30 mg	16 W	45	34 (73.9%)	5 (10.9%)	3 (6.5%)	N/A	1 (2.2%)	N/A	N/A
Igarashi A et al. [65]	Pediatric AD	30 mg	68 W	89	83 (93%)	15 (17%)	7 (8%)	N/A	15 (17%)	11 (12%)	5 (6%)
Ständer S et al. [54]	PN	30 mg	12 W	34	23 (68%)	7 (21%)	N/A	N/A	N/A	1 (2.9%)	2 (6%)
Kwatra SG et al. [53]	PN	30 mg 60 mg	16 W	183	112 (61.2%)	21 (11.5%)	N/A	7 (3.8%)	1 (0.5%)	1 (0.5%)	3 (1.6%)
Yokozeki H et al. [55]	PN	30 mg	68 W	112	103 (92%)	35 (31.2%)	8 (7.1%)	7 (6.3%)	6 (5.4%)	9 (8.0%)	7 (6.3%)
	PN	60 mg	68 W	113	103 (91.2%)	25 (22.1%)	10 (8.8%)	8 (7.1%)	8 (7.1%)	9 (8.0%)	7 (6.2%)

AD: atopic dermatitis; PN: prurigo nodularis; W: week; AEs: adverse events.

## Data Availability

The data relevant to this article may be requested from the corresponding author.

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
