# Peer review of "Cutaneous Adverse Events Following Nemolizumab Administration: A Review"

_jcm, 2025, doi:10.3390/jcm14093026_

Round 1
Reviewer 1 Report
Comments and Suggestions for Authors
This manuscript entitled "Cutaneous adverse events following nemolizumab administration: A review“ is very well written and researched, provides new and valuable information on effects of nemolizumab as a treatment option for pruritus, but it could be slightly improved by adding certain important elements as well as by editing some parts of the manuscript.
Although the article shows a very high level of research, I would advise to add more of your own comments and then to compare those to other authors.
The references are adequate, but ought to be written according to the standards of the journal, so please check for errors.
Author Response
#Reviewer1
Comment 1: This manuscript entitled "Cutaneous adverse events following nemolizumab administration: A review“ is very well written and researched, provides new and valuable information on effects of nemolizumab as a treatment option for pruritus, but it could be slightly improved by adding certain important elements as well as by editing some parts of the manuscript. Although the article shows a very high level of research, I would advise to add more of your own comments and then to compare those to other authors.
The references are adequate, but ought to be written according to the standards of the journal, so please check for errors.
Respond 1: Thank you for your valuable feedback. In response, we have added Figure 5 at the end of the Discussion section to present our treatment recommendations for cutaneous adverse events associated with nemolizumab. We also revised the Discussion to incorporate considerations on the management of these adverse events based on previous clinical trials and case series, thereby enhancing the overall quality and clarity of the section.
Additionally, to address the comments from other reviewers, we have expanded both the Introduction and the main body of Review part to provide a broader and more comprehensive overview of nemolizumab. Furthermore, all references have been revised in accordance with the journal’s formatting requirements as suggested.
We sincerely hope these revisions meet your expectations.
Thank you again for your thoughtful comments.
Reviewer 2 Report
Comments and Suggestions for Authors
Consider merging Sections 1.1 (AD) and 1.2 (PN) to avoid redundancy in describing shared pathophysiology (e.g., Th2 inflammation, barrier dysfunction).
Compare nemolizumab’s mechanism and AE profile with other biologics (e.g., dupilumab, lebrikizumab) to indicate its uniqueness.
Consolidate overlapping discussions on Th2/Th17 shifts (e.g., link mechanistic hypotheses directly to clinical observations).
Add a separate subsection on risk factors (e.g., genetic predisposition, comorbidities) for nemolizumab-associated AEs.
Clarify controversies (e.g., IL-31 inhibition enhancing vs. suppressing Th2 inflammation) using updated evidence (e.g., OSMRβ signaling pathways).
Consider proposing/adding clinical algorithms for managing severe AEs (e.g., dose adjustments, biologics switching criteria) in a table/flowchart format.
Discuss conflicting findings (e.g., transient TARC elevation vs. long-term Th2 suppression) and their implications.
Incorporate data on: long-term nemolizumab safety, fungal infection cases post-treatment, and comparative studies with dupilumab/lebrikizumab.
For Fig. 1, add legends, directional arrows, and annotations to distinguish hypotheses from validated pathways.
Standardize formatting (e.g., resolution timelines, treatment protocols) for all case figures, and include captions with key clinical details (e.g., dosing schedule, response duration).
Consider adding tables to summarize AE incidence, risk factors, and management strategies.
Reduce unnecessary self-citations and prioritize landmark studies instead.
Address cost-related treatment discontinuation in real-world settings.
Discuss ethical implications of limited access to diagnostic tests (e.g., patch testing) due to financial constraints.
Comments on the Quality of English LanguageSimplify complex sentences (e.g., replace passive voice with active voice).
Eliminate redundant descriptions (e.g., repeated FLG mutation explanations in Introduction and Review sections).
Ensure consistency in terminology.
Proofread.
Author Response
#Reviewer 2
Comment 1: Consider merging Sections 1.1 (AD) and 1.2 (PN) to avoid redundancy in describing shared pathophysiology (e.g., Th2 inflammation, barrier dysfunction).
Respon 1: Thank you for your valuable feedback. We have added a new subsection within the Introduction to provide a detailed description of cases of atopic dermatitis and prurigo nodularis.
Comment 2: Compare nemolizumab’s mechanism and AE profile with other biologics (e.g., dupilumab, lebrikizumab) to indicate its uniqueness.
Respond 2: Thank you for your valuable feedback. We have addressed the efficacy and adverse effects of both dupilumab and nemolizumab in detail in Section 1.3.
Comment 3: Consolidate overlapping discussions on Th2/Th17 shifts (e.g., link mechanistic hypotheses directly to clinical observations).
Respond 3: Thank you for your valuable feedback. As the content was partially repeated in the Review section, we created a new subsection (1.4) in the Introduction to clearly and concisely summarize the relationship between cytokine balance and nemolizumab. This structural change was made to enhance clarity and avoid redundancy.
Comment 4: Add a separate subsection on risk factors (e.g., genetic predisposition, comorbidities) for nemolizumab-associated AEs.
Respond 4: Thank you for your thoughtful comment. Currently, risk factors for cutaneous adverse events have not been thoroughly discussed in existing clinical trials or case series. Similarly, at our institution, the number of cases remains limited, and no significant risk factors have been identified to date. Therefore, it was difficult to include a detailed discussion of this topic in the current manuscript. Further accumulation of cases and dedicated research will be necessary to address this important issue in the future.
Comment 5: Clarify controversies (e.g., IL-31 inhibition enhancing vs. suppressing Th2 inflammation) using updated evidence (e.g., OSMRβ signaling pathways).
Respond 5: Thank you for your insightful comment. As nemolizumab has only been studied for a few years, unfortunately, there is currently no additional evidence beyond what is already included in the manuscript. Therefore, we have revised Section 1.4 to provide a more detailed and clear explanation of the relationship between IL-31 and oncostatin M (OSM), ensuring the content is conveyed in an accessible and comprehensive manner.
Comment 6: Consider proposing/adding clinical algorithms for managing severe AEs (e.g., dose adjustments, biologics switching criteria) in a table/flowchart format.
Respond 6: Thank you for your insightful comments. Upon reviewing previous clinical trials and published case series, we were unable to identify a standardized definition for what constitutes a “severe” adverse event (AE) in this context. In recent dermatological practice, the importance of shared decision-making has been increasingly emphasized. Therefore, we have added to the Discussion section the notion that the severity of AEs may not be solely determined by clinicians, but should also reflect the patient’s perspective—such as when skin symptoms worsen to the extent that the patient expresses a desire to modify or discontinue treatment. We also noted that, moving forward, more objective criteria—such as pruritus scores or the extent of skin involvement—may be developed to better define severe AEs in a standardized manner.
Comment 7: Discuss conflicting findings (e.g., transient TARC elevation vs. long-term Th2 suppression) and their implications.
Respond 7: Thank you for your insightful comment. We have added a detailed explanation to the newly created subsection 1.4, highlighting that the effects of nemolizumab on Th2-driven inflammation remain controversial. While some studies suggest it may have a suppressive effect, others indicate a potential for activation, and we have incorporated this complexity into the revised text.
Comment 8: Incorporate data on: long-term nemolizumab safety, fungal infection cases post-treatment, and comparative studies with dupilumab/lebrikizumab.
Respond 8: Thank you for your valuable feedback. We have addressed the efficacy and adverse effects of nemolizumab in Section 1.3 and at the beginning of the Review section. Although there have been no previously reported cases of fungal infections associated with nemolizumab, we have added a discussion on the potential influence of nemolizumab on fungal infections, based on the known association between dupilumab treatment and increased fungal infection risk.
The relationship between dupilumab and nemolizumab has been discussed in greater detail in Section 1.3. As lebrikizumab has not been approved for prurigo nodularis, we chose to focus the discussion on dupilumab and nemolizumab, and therefore did not include lebrikizumab in the main text.
Comment 9: For Fig. 1, add legends, directional arrows, and annotations to distinguish hypotheses from validated pathways.
Respond 9: Thank you for your valuable comment. We considered revising Figure 1; however, the adverse events depicted are all based on theoretical hypotheses, and no causal relationships have been definitively established. Additionally, the cytokine balance described remains speculative. Therefore, we have decided not to modify the figure, as incorporating the suggested elements would go beyond the scope of the current evidence.
Comment 10: Standardize formatting (e.g., resolution timelines, treatment protocols) for all case figures, and include captions with key clinical details (e.g., dosing schedule, response duration).
Respond 10: Thank you for your constructive feedback. We have standardized the descriptions of patient age, sex, underlying conditions, nemolizumab dosage, and symptom changes to ensure consistency throughout the manuscript. As this is a review article, we have presented these patient characteristics concisely to provide a clear understanding of the clinical context.
Comment 11: Consider adding tables to summarize AE incidence, risk factors, and management strategies.
Respond 11: Thank you for your valuable feedback. First, regarding the incidence of adverse events (AEs), we have summarized the reported side effects from previous clinical trials in a table at the beginning of the Review section. Second, for management strategies, we discussed findings from past clinical trials and published case series, and incorporated our own clinical experience to present our treatment suggestions in Figure 5.
Although we considered discussing risk factors for the development of cutaneous adverse events, there is currently very limited data available on this topic, and our institution has not yet accumulated a sufficient number of cases. Therefore, we regret that we were unable to include this aspect in the present review.
Comment 12: Reduce unnecessary self-citations and prioritize landmark studies instead.
Respond 12: Thank you for your feedback. We reduced self-citations, and instead cited clinical trials or fundamental research articles.
Comment 13: Address cost-related treatment discontinuation in real-world settings.
Respond 13: Thank you for your insightful comments. In the Discussion section, we have added a statement addressing the limitations some patients face in continuing treatment with nemolizumab due to financial constraints, as well as the challenges certain patients encounter in switching to alternatives such as dupilumab or lebrikizumab due to adverse events associated with nemolizumab.
Comment 14: Discuss ethical implications of limited access to diagnostic tests (e.g., patch testing) due to financial constraints.
Respond 14: Thank you for your thoughtful comment. While it is true that the inability to undergo allergy testing may potentially influence treatment decisions, in real-world clinical practice, such tests are not always reliable due to the possibility of false positives and false negatives. Therefore, we often choose not to perform allergy testing. Instead, we frequently rely on clinical course and other contextual factors to identify the causative agent and make decisions regarding drug substitution. For this reason, the absence of allergy testing may not necessarily have a significant impact on the overall treatment outcome. We have added this clarification to the Discussion section.
Comment 15: Simplify complex sentences (e.g., replace passive voice with active voice).
Respond 15: Thank you for your comment. We revised the complex sentences as you pointed out.
Comment 16: Eliminate redundant descriptions (e.g., repeated FLG mutation explanations in Introduction and Review sections).
Respond 16: you for your valuable comments. We have added a new section in the Introduction to provide detailed background information on atopic dermatitis and prurigo nodularis, and accordingly removed the redundant explanations from the Review section to improve clarity and avoid repetition.
Comment 17: Ensure consistency in terminology. Proofread.
Respond 17: Thank you for your valuable feedback. To ensure consistency in terminology and improve the overall quality of the manuscript, we have commissioned professional proofreading.
We sincerely hope these revisions meet your expectations.
Thank you again for your thoughtful comments.
Round 2
Reviewer 2 Report
Comments and Suggestions for Authors
Remove overlapping discussions on Th2 inflammation and barrier dysfunction in Sections 1.3 and 2.
Briefly mention lebrikizumab’s exclusion (e.g., “Lebrikizumab was omitted due to its lack of approval for PN”).
Consider adding a table comparing AE rates (e.g., conjunctivitis in dupilumab vs. eczema in nemolizumab).
Hypothesize potential risk factors (e.g., “Comorbid AD or FLG mutations may predispose patients to AEs”) based on case reports.
Add a supplementary table defining severity criteria (e.g., “Severe AE: ≥4-point increase in pruritus score or >30% BSA involvement”).
Include registry data on long-term AE rates if possible, including specific examples such as “post-marketing surveillance shows X% discontinuation due to fungal infections”.
Remove self-citations (e.g., Ref. 94, 95) unless critical to methodology.
Cite real-world data if possible (e.g., “X% of patients discontinued due to cost in [registry study]”).
Revise passive voice (e.g., “It has been reported” → “Studies report”).
Author Response
Dear reviewer 2,
Comment 1: Remove overlapping discussions on Th2 inflammation and barrier dysfunction in Sections 1.3 and 2.
Thank you for your feedback. We have revised and simplified the section in Chapter 2 related to cytokine balance accordingly.
Comment 2: Briefly mention lebrikizumab’s exclusion (e.g., “Lebrikizumab was omitted due to its lack of approval for PN”).
Respond 2: Thank you for your valuable feedback. As suggested, we have added a statement to Section 1.2 clarifying that this review focuses exclusively on dupilumab and nemolizumab, which are approved for both AD and PN, and that lebrikizumab and tralokinumab, which are approved only for AD, were excluded from the discussion.
Comment 3: Consider adding a table comparing AE rates (e.g., conjunctivitis in dupilumab vs. eczema in nemolizumab).
Respond 3: Thank you for your comment. We considered adding a table for dupilumab; however, since dupilumab's skin adverse events primarily concern refractory AD cases and do not typically include the urticaria, exacerbated eczema, or edematous erythema seen with nemolizumab, a direct comparison proved challenging. Therefore, we decided not to include such a table in this manuscript. Regarding conjunctivitis, its incidence is highly variable due to differences in eosinophil counts across clinical trials, and several studies have already comprehensively reviewed this topic. As our focus is on the cutaneous adverse events associated with nemolizumab, we have given only a brief mention of dupilumab-related events. We have added information regarding rare adverse events associated with dupilumab, including alopecia areata, arthritis, vitiligo, and rosacea-like folliculitis.
Comment 4: Hypothesize potential risk factors (e.g., “Comorbid AD or FLG mutations may predispose patients to AEs”) based on case reports.
Respond 4: Thank you for your comment. We have added a discussion point proposing that in AD or PN cases with pronounced erythema and barrier dysfunction—where baseline IL-4 and IL-13 levels are presumed to be high—nemolizumab may transiently elevate TARC levels and activate Th2 responses, potentially increasing the risk of Th2-related cutaneous AEs. In contrast, in cases characterized by marked lichenification or heightened pruritus sensitivity—where IL-31 is more central and IL-4/IL-13 levels are likely lower—such AEs may be less common. These hypotheses have been incorporated into the Discussion section. Further post-marketing case accumulation is needed to enable stratification and identification of high-risk groups.
Comment 5: Add a supplementary table defining severity criteria (e.g., “Severe AE: ≥4-point increase in pruritus score or >30% BSA involvement”).
Respond 5: Thank you for your feedback. As noted, clear severity classifications based on BSA involvement or pruritus worsening have not been established, even in clinical trials. At our institution, we adopt a patient-centered approach, classifying severity based on shared decision-making (SDM): whether the patient can continue treatment with rescue therapy, prefers to extend the dosing interval, or wishes to discontinue treatment. We have added a statement emphasizing the need for standardized criteria, such as BSA involvement and pruritus intensity, to guide future assessments.
Comment 6: Include registry data on long-term AE rates if possible, including specific examples such as “post-marketing surveillance shows X% discontinuation due to fungal infections”.
Thank you for your feedback. The main biologics used in atopic dermatitis are dupilumab, tralokinumab, and lebrikizumab, while nemolizumab is used less frequently and has only recently become available. As a result, there are very few studies outside of clinical trials that examine its long-term efficacy and safety. Therefore, we believe the 68-week data from clinical trials, as shown in Table 2, can serve as a substitute for discussing long-term adverse events.
Comment 7: Remove self-citations (e.g., Ref. 94, 95) unless critical to methodology.
Respond 7: Thank you for your feedback. We believe that our case report cited as reference 94 represents the first documented instance of erythema multiforme following nemolizumab administration, and we consider it essential for illustrating potential drug eruptions associated with nemolizumab. Additionally, we have replaced reference 80 with another publication that provides a more detailed explanation of our case.
Comment 8: Cite real-world data if possible (e.g., “X% of patients discontinued due to cost in [registry study]”).
Respond 8: Thank you for your insightful comment. We apologize, but data regarding the proportion of patients who discontinued nemolizumab treatment due to cost-related reasons were not available. As nemolizumab has only recently been introduced, further accumulation of clinical cases is needed moving forward.
Comment 9: Revise passive voice (e.g., “It has been reported” → “Studies report”).
Respond 9: Thank you for your feedback. We have revised the manuscript to replace the expression “~ has been reported” with alternative phrasings wherever possible.
The revised sections have been clearly marked in red for your reference. I would appreciate it if you could review them at your convenience. Thank you in advance.